Aberrant expression of two miRNAs promotes proliferation, hepatitis B virus amplification, migration and invasion of hepatocellular carcinoma cells: evidence from bioinformatic analysis and experimental validation

Liu Yanming 1 2
Cao Yue 3
Cai Wencan 2
Wu Liangyin 2
Zhao Pingsen 2
Liu Xin-guang 1 xgliu@gdmu.edu.cn
1 Guangdong Provincial Key Laboratory of Medical Molecular Diagnostics, Institute of Aging Research, Guangdong Medical University , Dongguan, Guangdong , China
2 Department of Clinical Laboratory, YueBei People’s Hospital , Shaoguan, Guangdong , China
3 Department of Medical Technology, Medical College of Shaoguan University , Shaogguan, Guangdong , China
Uversky Vladimir
Electronic publication date: 2020 Apr 29
Publication date: 2020
Volume: 8
Electronic Location ID: e9100
Received 2019 Dec 13; Accepted 2020 Apr 9
Copyright: © 2020 Liu et al.
Copyright year: 2020
Copyright holder: Liu et al.
License: This is an open access article distributed under the terms of the Creative Commons Attribution License, which permits unrestricted use, distribution, reproduction and adaptation in any medium and for any purpose provided that it is properly attributed. For attribution, the original author(s), title, publication source (PeerJ) and either DOI or URL of the article must be cited.
License URL: https://creativecommons.org/licenses/by/4.0/

Keywords: microRNA, Hepatocellular carcinoma, Bioinformatic analysis, Invasion, HBV amplification, Proliferation

Funding: National Natural Science Foundation of China 81671399 and 81971329 Guangdong Medical Research Fund B2018262 Shaoguan Health Bureau Research Fund Y19042 Shaoguan Science and Technology Fund 2018sn095 YueBei People’s Hospital Natural Science Foundation of Guangdong Province, China 2016A030307031 This work was supported by the National Natural Science Foundation of China (Nos. 81671399 and 81971329), Guangdong Medical Research Fund (No. B2018262), Shaoguan Health Bureau Research Fund (No. Y19042), Shaoguan Science and Technology Fund (No. 2018sn095) and Outstanding Research Fund of YueBei People’s Hospital, Natural Science Foundation of Guangdong Province, China (No. 2016A030307031). The funders had no role in study design, data collection and analysis, decision to publish, or preparation of the manuscript.

==============================
Background

As key negative regulators of gene expression, microRNAs (miRNAs) play an important role in the onset and progression of hepatocellular carcinoma (HCC). This study aimed to identify the miRNAs involved in HCC carcinogenesis and their regulated genes.

Methods

The Gene Expression Omnibus (GEO) dataset (GSE108724) was chosen and explored to identify differentially expressed miRNAs using GEO2R. For the prediction of potential miRNA target genes, the miRTarBase was explored. Enrichment analysis of Gene Ontology (GO) and the Kyoto Encyclopedia of Genes and Genomes (KEGG) was performed by the DAVID online tool. The hub genes were screened out using the CytoHubba plug-in ranked by degrees. The networks between miRNAs and hub genes were constructed by Cytoscape software. MiRNA mimics and negative control were transfected into HCC cell lines and their effects on proliferation, hepatitis B virus DNA (HBV-DNA) replication, TP53 expression, migration, and invasion were investigated. The following methods were employed: MTT assay, quantitative PCR (qPCR) assay, western blotting, wound healing assay, and transwell assay.

Results

A total of 50 differentially expressed miRNAs were identified, including 20 upregulated and 30 downregulated miRNAs, in HCC tumor tissues compared to matched adjacent tumor-free tissues. The top three upregulated (miR-221-3p, miR-222-3p, and miR-18-5p) and downregulated (miR-375, miR-214-3p and miR-378d) miRNAs, ranked by |log2 fold change (log2FC)|, were chosen and their potential target genes were predicted. Two gene sets, targeted by the upregulated and the downregulated miRNAs, were identified respectively. GO and KEGG pathway analysis showed that the predicted target genes of upregulated and downregulated miRNAs were mainly enriched in the cell cycle and cancer-related pathways. The top ten hub nodes of gene sets ranked by degrees were identified as hub genes. Analysis of miRNA-hub gene network showed that miR-221-3p and miR-375 modulated most of the hub genes, especially involving regulation of TP53. The q-PCR results showed that miR-221-3p and miR-375 were markedly upregulated and downregulated, respectively, in HCC cells and HCC clinical tissue samples compared to non-tumoral tissues. Furthermore, miR-221-3p overexpression significantly enhanced proliferation, HBV-DNA replication, as well as the migration and invasion of HCC cells, whereas miR-375 overexpression resulted in opposite effects. Western blotting analysis showed that the overexpression of miR-221-3p and miR-375 reduced and increased TP53 expression, respectively.

Conclusion

The present study revealed that miR-211-3p and miR-375 may exert vital effects on cell proliferation, HBV-DNA replication, cell migration, and invasion through the regulation of TP53 expression in HCC.

Introduction

With the second-highest morbidity, third-highest mortality rate, and fifth-highest global incidence in China, hepatocellular carcinoma (HCC) is one of the most lethal malignant tumors, and new cases are increasing worldwide each year (Yanming et al., 2019). The risk factors for HCC are mainly represented by chronic hepatitis B virus (HBV) or hepatitis C virus (HCV) infection, cirrhosis, and alcoholic liver disease (Xie et al., 2019; Zhang et al., 2019). Alpha-fetoprotein (AFP), the most important tumor marker for HCC diagnosis, has been used for decades but results in a clinically unsatisfactory sensitivity of 40–65% and a specificity of 76–96% (Yanming et al., 2019). Therefore, the identification of novel biomarkers of HBV replication and HCC proliferation, migration, and invasion is urgent and crucial for developing effective diagnostic, therapeutic, and prognostic follow-up strategies. MicroRNAs (miRNAs) are endogenous 21–25 nt RNAs that induce mRNA degradation or inhibition of translation by binding to the 3′-untranslated region of target messenger RNAs (mRNAs). It has been demonstrated that the abnormal expression of specific miRNAs promotes the proliferation, invasion, and metastasis of tumor cells (Wang et al., 2018). Therefore, thorough investigation of miRNAs that are functionally related to HBV-infection, as well as to the proliferation, migration, and invasion of HCC cells is of paramount importance.

High-throughput microarray technology has been extensively applied for decades owing to its high sensitivity and processivity. Bioinformatic analysis is widely used to identify differentially expressed miRNAs (DE-miRNAs) and functional pathways involved in carcinogenesis and HCC progression. The combination of these two approaches may help understand the pathogenesis of tumors and improve early diagnosis, treatment, and prognosis of HCC patients (Mantione et al., 2014).

In the present study, the GSE108724 dataset, comprising seven pairs of HCC and matched adjacent tumor-free tissues, was applied to the screening of 50 differentially expressed miRNAs (DE-miRNAs), including 20 upregulated and 30 downregulated miRNAs related to HCC pathogenesis. The top three upregulated and downregulated miRNAs, ranked by |log2 fold change (log2FC)|, were selected to predict potential target genes, respectively. Enriched Gene Ontology (GO) terms and Kyoto Encyclopedia of Genes and Genomes (KEGG) pathways of each gene set were identified using the DAVID online tool. Protein–protein interaction (PPI) networks were constructed to identify the top ten hub nodes of each gene set, ranked by degrees as hub genes, using the CytoHubba plug-in. Then, the networks established by the top three upregulated and downregulated miRNAs with their 10 target hub genes were constructed. MiR-221-3p and miR-375 exhibited the most regulatory relationships with their hub genes, with special regard to TP53 regulation. Subsequently, in vitro validation experiments were performed. As the novelty and innovation of this present research, bioinformatics and experimental validation were combined to explore the effects of miR-211-3p and miR-375 on HBV DNA amplification, TP53 expression, as well as on the proliferation, migration, and invasion of HCC cell lines.

Materials and Methods

Microarray data

To explore the role of specific miRNAs in HCC pathogenesis, the GSE108724 dataset (Zhu et al., 2019) was selected and downloaded from GEO (https://www.ncbi.nlm.nih.gov/geo/). This dataset, based on the GPL20712 platform (Agilent-070156 Human miRNA), contained miRNA expression profiles of 7 pairs of HCC and matched adjacent tumor-free tissues.

DE-miRNA identification and target gene prediction

The DE-miRNAs between HCC and matched adjacent tumor-free tissues were screened using GEO2R (http://www.ncbi.nlm.nih.gov/geo/geo2r) according to the cut-off criterion that P-value < 0.05 and |log2 fold change (log2FC)| > 2, an online interactive web tool with built-in GEOquery and limma R packages that allows to compare two or more datasets across experimental conditions. The GEOquery R package parses GEO data into R data structures that can be used by other R packages (Davis & Meltzer, 2007). The Limma R package is used to homogenize microarray data and to reduce the occurrence of false positives by applying multiple-testing corrections on P-values (Ritchie et al., 2015). The miRTarBase online dataset (http://mirtarbase.mbc.nctu.edu.tw/php/index.php) (version 7.0), containing extensive information on experimentally validated miRNA-target interactions, was utilized for predicting potential miRNA target genes (Chou et al., 2018). Only the top three upregulated and downregulated miRNAs, ranked by |log2FC |, were enrolled for target gene prediction analysis and formed two new gene sets of upregulated and downregulated miRNAs, on which GO, KEGG, and hub gene identification analyses were further performed.

GO, KEGG, hub gene identification and miRNA-hub gene network construction

The Database for Annotation, Visualization and Integrated Discovery (DAVID; https://david.ncifcrf.gov) (version 6.8) (Da Huang, Sherman & Lempicki, 2009a, 2009b), providing a comprehensive set of functional annotation information including large lists of genes, allowing for the retrieval of biological information, was applied to perform functional annotation and pathway enrichment analysis for the two gene sets, including GO (Edgar, Domrachev & Lash, 2002) and KEGG (Kanehisa et al., 2017) pathway analysis. Search Tool for the Retrieval of Interacting Genes (STRING) (https://string-db.org/) is the most popular database of known and predicted protein–protein interactions (PPI). The PPI networks of the two gene sets were constructed using STRING (version 11.0), and interactions with a combined score >0.4 were considered statistically significant (Szklarczyk et al., 2019). The top ten hub genes of the two gene sets ranked by degrees were selected by using the CytoHubba plug-in (http://apps.cytoscape.org/apps/cytohubba) (version 0.1) (Chin et al., 2014). The Cytoscape software (https://cytoscape.org) (version 3.7.2) was used to construct the networks between three miRNAs and their respective top ten hub target genes (Shannon et al., 2003). P-value < 0.05 was considered as statistically significant.

Analysis of gene and miRNA expression, survival, and co-expression of miRNAs-hub genes

The Cancer Genome Atlas (TCGA) analysis tool of the UALCAN database (http://ualcan.path.uab.edu/), an interactive web resource for the analysis of cancer omics data, was utilized to explore differences in the expression of hub genes between normal and tumor tissue in patients with liver hepatocellular carcinoma (LIHC) (Chandrashekar et al., 2017). The OncomiR database (http://www.oncomir.org/), an online resource for the evaluation of cancer-related miRNA dysregulation, was applied for performing Kaplan–Meier survival analysis based on the expression level of miR-221-3p and miR-375, with 50% as percentile cutoff value in HCC (Wong et al., 2018). The Pan-Cancer analysis tool of starBase (version 3.0) (http://starbase.sysu.edu.cn/panMirCoExp.php), identifying more than 2.5 million miRNA-mRNA interactions from multi-dimensional sequencing data, was used for miRNA-target gene co-expression analysis (Li et al., 2014). P-value of < 0.05 was considered as statistically significant.

HCC cell lines, clinical samples, and cell transfection

Immortalized normal human liver HL7702 cells and the HCC cell lines HepG2, HepG2.2.15, and MHCC-LM3 were purchased from Cellcook Co., Ltd. (Guangzhou, China). All cells were cultured in Dulbecco’s modified Eagle’s medium/High Glucose (DMEM: 8885329; Gibco, Shanghai, China) with 10% fetal bovine serum (FBS: 11011-8611; Tianhang, Hangzhou, China) and maintained under a humidified atmosphere of 5% CO2 at 37 °C. With the written informed consent from participants and the approval obtained from the institutional ethics review board of YueBei people’s Hospital (approval number: SUMC-IRB-2017), HCC tumor tissues and matched adjacent tumor-free tissues were obtained from YueBei People’s Hospital. According to the manufacturer’s instructions, the riboFECT CP Transfection Kit (C10511-05; RiboBio, Guangzhou, China) was used to transfect miRNA mimics and negative control (RiboBio, Guangzhou, China) into HCC cell lines.

RNA isolation and quantitative reverse transcription PCR

Total RNA from different cell lines and human tissues was extracted using the RNA Extraction Kit (R1405; GBCBIO, Guangzhou, China). The extracted RNA was stored at −80 °C after measuring the concentration and quality. Stem-loop reverse transcriptase (RT) primers for two miRNAs were designed and purchased from TSINGKE Biological Technology Co., Ltd., Beijing, China. Q-PCR was performed three times using SYBR Premix Ex Taq (Q111-02; Vazyme, Nanjing, China) and normalized with U6 small nuclear RNA as internal reference. The 2−ΔΔCT method was applied to determine the relative expression level of miRNAs. The sequences of qRT-PCR primers are shown in Table S1.

MTT cell proliferation assay

HepG2 cells (2 × 103 cells/well) were plated in 96-well plates. After transfection (final transfection concentration of miRNA mimics and negative control: 50 nM) and growth to 95% confluence, HepG2 cells were harvested. Cell proliferation was analyzed in triplicate by using the MTT assay kit (G7420-2; GBCBIO, Guangzhou, China) according to the manufacturer’s instruction.

HBV-DNA qPCR detection

HepG2.2.15 cells (1 × 105 cells/well) were seeded in 6-well plates. After transfection, the HBV-DNA quantitative kit (2019005; DAAN Gene, Guangzhou, China), including the reagents for DNA extraction, PCR amplification, calibration, and quality control, was used for measuring the level of HBV-DNA in one mL of cell culture supernatant, as per manufacturer’s instructions.

Western blot

Western blot was performed as previously described with minor modifications (Marangos, 2016). HepG2 cells (1 × 105 cells/well) were seeded in 6-well plates. After transfection and growth to 95% confluence, the cells were harvested. The following primary antibodies were used: anti-p53 (ab131442; Abcam, Cambridge, UK, 1:100) and anti-GAPDH (EPR16891; Abcam, Cambridge, UK, 1:100). The horseradish peroxidase-labeled secondary antibody ab205718, Abcam, Cambridge, UK was used (1:500). A DAB Horseradish Peroxidase Color Development Kit (G3433; GBCBIO, Guangzhou, China) was employed to visualize p53 and GAPDH. The colored bands on the PVDF membrane were photographed with an OLYMPUS microscope (Tokyo, Japan) and permanently stored.

Wound healing assays

HepG2 cells (1 × 105 cells/well) were seeded in 12-well plates. After transfection and growth to 95% confluence, a sterilized pipette tip was used to scratch a wound. The wounded areas were photographed with a microscope, immediately and after 24 h (magnification: 100×).

Transwell assay

The cell invasion assay was performed using 24-well transwell chambers (3422; Corning Inc., New York, USA). The transwell chamber (upper compartment) was separated from the 24-well plate (lower compartment) by polycarbonate membrane with a pore size of 8 μm. The upper surface of the polycarbonate membrane was coated with Matrigel (354230; BD Bioscience, San Jose, CA, USA, 1:8 dilution). After transfection, 200 μL of serum-free medium containing 1 × 105 HepG2 cells were added to the upper compartment. Next, 600 μL of medium containing 20% FBS were added to the lower compartment. Then, after a 24-h incubation at 37 °C and removal of the cells on the upper surface of the membrane in the upper compartment using a cotton bud, the membranes were fixed (95% methanol) for 30 min, stained (0.1% crystal violet) for 20 min at room temperature, and three visual fields were photographed (magnification: 200×) under a microscope (OLYMPUS, Tokyo, Japan). The experiments were independently repeated in triplicate.

Statistical analysis

The results were expressed as mean±SD. The raw data obtained with the HBV-DNA assay were subjected to base 10 logarithmic conversion before statistical analysis. One-way ANOVA analysis was used to compare the miRNA expression between four cell lines. The two-independent t-test was used to compare miRNAs expression between para-tumor and tumor tissues. Unless otherwise noted, statistical diagrams were drawn using GraphPad Prism (version 8.0.1; GraphPad Software, San Diego, CA, USA). For all analyses, P < 0.05 was regarded as statistically significant.

Results

Identification of DE-miRNAs and of their potential target genes in HCC

The GEO2R online tool was applied to screen 50 DE-miRNAs between 7 HCC tissues and their matched adjacent tumor-free tissues in the GSE108724 dataset (P-value < 0.05 and |log2FC| > 2). Twenty upregulated and thirty downregulated DE-miRNAs are shown in the volcano plot (Fig. 1). Tables 1 and 2 show the top ten upregulated and downregulated DE-miRNAs, respectively, ranked by |log2FC|. A total of 741 potential target genes were predicted for the top three upregulated miRNAs (miR-221-3p, and miR-222-3p, and miR-18b-5p), while 696 genes were identified as candidate targets of the top three downregulated miRNAs (miR-375, miR-214-3p, and miR-378d) by using the miRTarBase online tool.

Figure 1 Volcano plot of DE-miRNAs between seven pairs of HCC and matched adjacent tumor-free tissues in the GSE108724 dataset.

Red and green dots represent upregulated and downregulated miRNAs, respectively, based on P-value < 0.05 and |log2FC| > 2; black dots represent miRNAs with no significant expression differences.

Table 1 The top ten upregulated miRNAs between HCC and matched adjacent tumor-free tissues ranked by |log2FC|.

ID	log2FC	P-value	t	B	
hsa-miR-221-3p	3.75	1.27E−03	4.10	−0.73	
hsa-miR-222-3p	3.57	1.70E−03	3.95	−0.99	
hsa-miR-18b-5p	3.41	1.12E−02	2.96	−2.70	
hsa-miR-500a-3p	2.80	2.57E−02	2.52	−3.44	
hsa-miR-196b-5p	2.80	1.62E−02	2.77	−3.03	
hsa-miR-7-5p	2.60	1.21E−02	2.92	−2.77	
hsa-miR-6516-3p	2.55	2.39E−02	2.56	−3.38	
hsa-miR-34a-3p	2.46	1.72E−02	2.73	−3.09	
hsa-miR-362-3p	2.13	1.08E−02	2.97	−2.67	
hsa-miR-339-3p	2.02	1.35E−02	2.86	−2.87	

Table 2 The top ten downregulated miRNAs between HCC and matched adjacent tumor-free tissues ranked by |log2FC|.

ID	Log2FC	P value	t	B	
hsa-miR-375	−6.41	1.51E−07	−10.22	6.72	
hsa-miR-214-3p	−4.90	1.94E−03	−3.88	−1.11	
hsa-miR-378d	−4.20	2.38E−03	−3.77	−1.30	
hsa-miR-486-5p	−4.13	1.37E−03	−4.06	−0.80	
hsa-miR-199a-5p	−3.86	3.51E−03	−3.56	−1.65	
hsa-miR-8089	−3.54	1.34E−03	−4.08	−0.77	
hsa-miR-338-3p	−3.28	2.91E−03	−3.66	−1.48	
hsa-miR-7845-5p	−3.20	1.98E−03	−3.87	−1.13	
hsa-miR-199a-3p	−2.95	4.74E−04	−4.64	0.16	
hsa-miR-10a-5p	−2.13	1.23E−03	−4.12	−0.70	

Functional enrichment analysis of target genes

To comprehensively investigate the biological functions of target genes, GO and KEGG pathway enrichment analyses were carried out using the DAVID online tool. GO analysis can be divided into three functional groups: biological process group (BP), molecular function group (MF), and cellular component group (CC). GO analysis of the 741 potential target genes of the three upregulated miRNAs revealed that changes in BP were mainly enriched in negative regulation of transcription from RNA polymerase II promoter, positive regulation of transcription, and cell-cell adhesion (Fig. S1A); changes in MF were mainly enriched in poly(A) RNA binding, protein binding, and cadherin binding involved in cell–cell adhesion (Fig. S1C); changes in CC were mainly enriched in nucleoplasm, nucleus, and cytosol (Fig. S1E). GO analysis of the 696 target genes of the three downregulated miRNAs revealed that changes in BP were mainly involved positive regulation of neuron apoptotic process, organ morphogenesis, and blood vessel remodeling (Fig. S1B); changes in MF were mainly involved GDP binding, transcription coactivator activity, and mRNA binding (Fig. S1D); changes in CC were mainly involved nucleus, extracellular exosome, and nucleoplasm (Fig. S1F). KEGG pathway analysis showed that the 741 candidate target genes of the three upregulated miRNAs were mainly enriched in cell cycle, cancer-related pathways, and FoxO signaling pathway (Fig. S1G); the 696 target genes of the three downregulated miRNAs were mainly involved cancer-related pathways, breast cancer, hepatitis B, and MAPK signaling pathway (Fig. S1H).

Hub gene identification, miRNA-hub gene network construction, and targeted gene expression analysis in HCC tissues

The top 10 nodes ranked by degrees of the CytoHubba plug-in of Cytoscape software were recognized as hub genes. Six (TP53, MYC, HSP90AA1, PTEN, CASP3, CTNNB1) of ten hub genes were in common between upregulated and downregulated DE-miRNAs, albeit they exhibited different scores (Table 3). TP53, exhibiting the highest score among upregulated and downregulated DE-miRNAs, could play a vital role in HCC carcinogenesis or progression. The analysis of the miRNA-Hub gene network revealed that miR-221-3p potentially modulated seven (PTEN, TP53, ACTB, ESR1, CTNNB1, CASP3, and UBC) of ten hub genes. Moreover, miR-222-3p and miR-18b-5p regulated six and two hub genes, respectively (Fig. 2A). Interestingly, miR-375 potentially modulated five (HSP90AA1, CASP3, MYC, TP53, and CTNNB1) of 10 hub genes. MiR-214-3p and miR-378d could regulate three and zero hub genes, respectively (Fig. 2B). Eleven significantly upregulated hub genes (UBC, ACTB, TP53, HSP90AA1, PTEN, CASP3, CTNNB1, AKT1, MAPK3, CDC42, and MAPK1) (Figs. 3A, 3D, 3E, 3I–3P) and two significantly downregulated hub genes (ESR1 and MYC) (Figs. 3C and 3H) were compared in HCC tissues and normal liver tissues using the TCGA analysis tool of the UALCAN database. The results of TCGA analysis showed that the expression level of TP53 was positively related to HCC tumor stage and metastasis (Figs. 3F and 3G). TP53, one of the hub genes in common between the two sets of DE-miRNAs, exhibiting the highest score, maybe the most potentially hub gene of miR-221-3p and miR-375. Based on the above findings, we speculated that miR-221-3p and miR-375 might play critical roles in HCC by regulating TP53 expression.

Table 3 The top ten hub genes of target gene sets predicted by three upregulated and downregulated miRNAs ranked by degrees.

Upregulated DE-miRNAs	Downregulated DE-miRNAs	
Rank	Name	Score	Rank	Name	Score	
1	TP53	172	1	TP53	151	
2	MYC	128	2	AKT1	141	
3	HSP90AA1	108	3	MYC	107	
3	UBC	105	4	PTEN	90	
5	PTEN	104	5	MAPK3	87	
6	CCND1	91	6	HSP90AA1	84	
7	CASP3	82	7	CTNNB1	83	
8	CTNNB1	82	8	CDC42	82	
9	ESR1	81	9	MAPK1	77	
10	ACTB	79	10	CASP3	76	

Figure 2 The miRNA-hub gene network was constructed using Cytoscape software.

(A) Network of three upregulated miRNAs and their predicted hub genes. (B) Network of three downregulated miRNAs and their predicted hub genes. Red diamonds represent upregulated miRNAs, green diamonds represent downregulated miRNAs, cyan rectangles represent hub genes, lines represent the interaction between miRNAs and hub genes, and free diamonds and rectangle represent the absence of interaction between miRNAs and hub genes.

Figure 3 mRNA expression of the predicted hub genes of miR-221-3p and miR-375 from the UALCAN LIHC database.

The hub genes were potential predictive targets of miR-211-3p (A–L) and miR-375 (E–P). (A) UBC expression based on sample type. (B) CCND1 expression based on sample type. (C) ESR1 expression base on sample types. (D) ACTB expression base on sample types. (E) TP53 expression base on sample types. (F) TP53 expression based on tumor grade. (G) TP53 expression based on nodal metastasis status. (H) MYC expression based on sample type. (I) HSP90AA1 expression based on sample type. (J) PTEN expression based on sample type. (K) CASP3 expression based on sample type. (L) CTNNB1 expression based on sample type. (M) AKT1 expression based on sample type. (N) MAPK3 expression based on sample type. (O) CDC42 expression based on sample type. (P) MAPK1 expression based on sample type. *P < 0.05; **P < 0.01; ***P < 0.001; ****P < 0.0001.

The expression level of miR-221-3p and miR‑375 in HCC cells and clinical samples

Three HCC cell lines (HepG2, HepG2.2.15, and HCC-LM3) and a normal liver cell line (HL7702) were employed to evaluate the expression level of miR-221-3p and miR-375. qRT-PCR revealed that the two miRNAs were significantly upregulated and downregulated in all three HCC cell lines, respectively, compared to HL7702 cells (Figs. 4A and 4D). Notably, the expression levels of miR-221-3p and miR-375 were the highest and lowest in HCC-LM3 compared to the other two HCC cell lines, respectively. Due to the high invasion capacity of HCC-LM3 cells (Zha et al., 2018), this suggested that miR-221-3p and miR-375 could be critical regulators of HCC cell invasion. HepG2.2.15 is an HCC cell line that carries and secretes HBV particles. Notably, in these cells, miR-221-3p expression was much higher, and that of miR-375 much lower, compared to HepG2 cells. These findings were consistent with an important role of miR-221-3p and miR-375 in the tumorigenesis of HBV-related HCC. Subsequently, qRT-PCR analysis of HCC tissue samples demonstrated that miR-221-3p and miR-375 were, respectively, upregulated and downregulated in 20 HCC tissue samples compared to their para-tumor tissues (Figs. 4B and 4E). Next, the OncomiR online database was employed to further evaluate the expression level and the prognostic value of miR-221-3p and miR-375 in HCC patients. The mean expression level of miR-221-3p was much higher, and that of miR-375 much lower, in HCC tissues compared to normal liver tissues (Figs. 4C and 4F). Interestingly, high miR-221-3p expression and low miR-375 expression tended to be correlated with poor overall survival. However, this association was not statistically significant (P = 0.06034 for miR-221-3p and P = 0.3215 for miR-375) (Figs. 4G and 4H). In conclusion, although further experimental validation is needed, miR-221-3p and miR-375 are potential prognostic biomarkers, with special regard to HBV amplification and HCC invasion.

Figure 4 The expression and prognostic roles of miR-221-3p and miR-375 in HCC.

The expression level of miR-221-3p (A) and miR-375 (D) in three HCC cell lines (HepG2, HepG2.2.15, MHCC-LM3) and one normal liver cell line (HL7702) detected by qRT-PCR. MiR-221-3p and miR-375 were significantly upregulated and downregulated in all three HCC cell lines, respectively, compared to HL7702 cells. The expression level of miR-221-3p (B) and miR-375 (E) in 20 clinical HCC samples vs matched para-tumor tissues detected by qRT-PCR. MiR-221-3p and miR-375 were, respectively, upregulated and downregulated in 20 HCC tissue samples compared to their para-tumor tissues. The mean expression level of miR-221-3p (C) and miR-375 (F) in multiple clinical HCC samples vs normal tissues, as assessed by OncomiR online analysis. The mean expression level of miR-221-3p and miR-375 in multiple clinical HCC samples was much higher and lower, respectively, than that of normal tissues. Kaplan–Meier survival curve of miR-221-3p (G) and miR-375 (H) based on percentile cutoff value 50% in HCC, as determined by OncomiR online analysis. high miR-221-3p expression and low miR-375 expression tended to be correlated with poor overall survival. However, this association was not statistically significant (P = 0.06034 for miR-221-3p and P = 0.3215 for miR-375). The cohort labeled as high-risk (red line) represents levels of miRNA expression above the average level; the cohort labeled as low-risk (blue line) represents miRNA expression levels below the average level. One-way ANOVA analysis was used to compare the miRNA expression between four cell lines. The two-independent t-test was used to compare miRNAs expression between para-tumor and tumor tissues. *P < 0.05; ***P < 0.001; ****P < 0.0001.

The effects of miR-221-3p and miR-375 overexpression on proliferation, HBV-DNA replication, migration, and invasion of HCC cells

HepG2 cells were transfected with 50 nM or 100 nM miRNA mimics. qRT-PCR analysis showed that miRNA expression increased with the concentration of transfected miRNA mimics (Fig. 5A). After experimental verification, 50 nM was selected as the most suitable concentration of the two miRNAs for subsequent transfections. First, we performed an MTT assay to explore the impact of miR-221-3p and miR-375 on HCC cell proliferation. The results showed that miR-221-3p and miR-375 overexpression significantly enhanced and decreased, respectively, HepG2 cell proliferation when compared to HepG2 cells transfected with the negative control (Fig. 5B). In addition, we performed an HBV-DNA qPCR assay to determine the effect of miR-221-3p and miR-375 overexpression on the replication of HBV-DNA. The qPCR results showed that the upregulation of miR-221-3p and miR-375 increased and decreased, respectively, the level of HBV-DNA in HepG2.2.15 cells (Fig. 5C). Moreover, western blot analysis indicated that miR-221-3p and miR-375 overexpression weakened and enhanced, respectively, the expression of TP53 (Fig. 5D). The co-expression experiments showed that the level of TP53 expression was strongly correlated with those of the two miRNAs in HCC patients (miR-221-3p: r = 0.313, P-value: 7.16E−10; miR-375: r = 0.036, P-value: 4.59E−1) (Figs. 5E and 5F). Moreover, we performed a wound healing assay to evaluate the role of miR-221-3p and miR-375 in HCC cell migration. HepG2 cell migration was positively correlated with miR-221-3p overexpression and negatively correlated with miR-375 overexpression (Figs. 6A and 6B). Finally, the effects of the two miRNAs on HCC cell invasion were verified by a transwell assay, which demonstrated that miR-221-3p and miR-375 overexpression enhanced and inhibited, respectively, HepG2 cell invasion (Figs. 6C and 6D). In summary, the data confirmed that miR-221-3p and miR-375 are potential regulators of TP53 expression, and could therefore affect HBV-DNA replication, as well as HCC cell proliferation, migration, and invasion.

Figure 5 Overexpression of miR-221-3p and miR-375 affect cell proliferation, HBV-DNA replication, and TP53 expression in HCC cell lines.

(A) The relative expression level of miR-221-3p/NC and miR-375/NC after transfection with NC, miR-221-3p mimics, and miR-375 mimics (at the final concentrations of 50 nM or 100 nM) in HepG2 cells. The miRNA mimics (miR-211-3p and miR-375) significantly increased the expression of the corresponding miRNAs at both concentrations. (B) The overexpression of miR-221-3p and miR-375 in HepG2 cells enhanced and reduced, respectively, cell proliferation, as assessed by MTT analysis. (C) The overexpression of miR-221-3p and miR-375 in Hepg2.2.15 cells increased and decreased, respectively, the level HBV-DNA, as demonstrated by qRT-PCR. (D) The overexpression of miR-221-3p and miR-375 downregulated and upregulated, respectively, TP53 expression, as assessed by western blotting. (E) and (F) Analysis of the co-expression of miR-211-3p/TP53 (E) and miR-375/TP53 (F) in HCC clinical tissue samples using starBase online software. The expression of both miR-221-3p and miR-375 was positively correlated with TP53 expression (r = 0.313 and P-value = 7.16E−10 for miR-221-3p; r = 0.036 and P-value = 0.495 for miR-375). The two-independent t-test was used to compare the difference of miRNA relative expression, MTT, and HBV-DNA. **P < 0.01; ***P < 0.001.

Figure 6 Overexpression of miR-221-3p and miR-375 enhanced and suppressed, respectively, the migration and invasion of HepG2 cell lines.

(A) The wound healing assay was performed after transfection with NC, miR-221-3p mimic (50 nM), or miR-375 mimic (50 nM), and an additional 24 h of culture (picture magnification: 100×; black bar: 100 μm); HepG2 transfected with miR-221-3p and miR-375 mimics exhibited higher and lower migration ability compared to NC-transfected cells, respectively. (B) Quantification from A. (C) The transwell assay was performed after transfection with NC, miR-221-3p mimic (50 nM) or miR-375 mimic (50 nM) and an additional 24 h of culture (picture magnification: 200×; black bar: 50 μm). HepG2 transfected with miR-221-3p miR-375 mimics were more and less invasive compared to those transfected with NC, respectively. (D) Quantification from C. The two-independent t-test was used to compare the difference of wound healing rate and cell number. **P < 0.01; ***P < 0.001.

Discussion

The pathogenesis HCC is still unclear, and the known etiological factors mainly include exogenous factors, such as hepatitis virus infection, and endogenous factors, such as excessive drinking, metabolic disorders, and environmental factors (Kokudo et al., 2019). The 5-year survival of HCC patients can be higher than 50% after early diagnosis (Forner, Llovet & Bruix, 2012). However, due to delayed diagnosis, most HCC patients are not suitable for further treatment and often have a poor prognosis.

Numerous studies had demonstrated that miRNAs play an important role in various biological processes of HCC cells, such as cell proliferation, HBV persistent infection, tumor suppressor genes expression, and metastasis. In China, the persistent exacerbation of HBV amplification is the main determinant of HCC recurrence (Yanming et al., 2019). A recent study showed that miR-146a SNP (C/G) is associated with the susceptibility to HBV infection and with spontaneous viral clearance (Khanizadeh et al., 2019). TP53 mutation, the most common mutation in HCC, affects the progression and prognosis of HCC (Long et al., 2019). Interestingly, miR-487a stimulates cancer cell proliferation via PIK3R1-mediated AKT signaling and promotes metastasis via SPRED2-induced mitogen-activated protein kinase signaling in HCC (Chang et al., 2017). Hence, it is urgent need to systematically analyzed and experimentally verified the role of specific miRNAs in HBV amplification, as well as in the proliferation, migration, and invasion of HCC cells.

In this study, a data-driven decision was applied to gradually narrow down the scope of experimental verification miRNAs (Lou et al., 2018). First, we screened 20 upregulated and 30 downregulated miRNAs on the basis of the miRNA expression profiles of the GSE108724 dataset. Subsequently, potential miRNA target genes were predicted for the top three upregulated and downregulated miRNAs ranked by |log2FC|. According to the results of GO and KEGG analysis, the identified miRNAs were implicated in focal adhesion, cell cycle regulation, HBV infection, and mitosis. Next, the top ten hub genes, ranked by degrees, of upregulated and downregulated miRNAs were identified. The generated miRNA-hub gene network showed that miR-221-3p and miR-375 modulated most of the hub genes. Moreover, the two latter miRNAs exhibited the highest score regarding TP53 regulation. However, as far as we know, the effect of TP53 expression modulated by miR-221-3p and miR-375 on the proliferation, invasion, metastasis, and HBV replication in HCC have rare been reported. It is of great significance to clarify the mechanism how the two miRNAs regulate the above phenotypes of HCC.

The present study confirmed that the expression of miR-221-3p and miR-375 was significantly upregulated and downregulated, respectively, in both HCC cell lines and HCC tissue samples. The upregulation of miR-212-3p and miR-375 enhanced and inhibited, respectively, HBV-DNA replication and amplification, as well as HCC cell migration and invasion. In addition, the upregulation of miR-212-3p and miR-375 exerted opposite effects on TP53 expression, respectively suppressing and promoting its transcription. Recent studies have demonstrated that miR-221-3p functions as an oncogene in specific types of human cancer, including HCC (Wang et al., 2019), non-small cell lung cancer (Yin, Zhang & Li, 2019), glioma (Milani et al., 2019), gastric cancer (Zhang et al., 2019) and breast cancer (Deng et al., 2017). It was also reported that miR-375 inhibits the proliferation and migration of many cancer cells (Guo et al., 2018; Shen et al., 2014; Wu et al., 2017). Moreover, miR-221 promotes HCC cell migration by regulating the expression of plant homeodomain finger 2 (PHF2), which was demonstrated to be a target gene of miR-221 (Fu et al., 2019). Furthermore, PHF2 acts as a vital regulator of cancer development in association with TP53, and is required for TP53-mediated cell death (Lee et al., 2015). Several studies demonstrated that miR-375 negatively regulates epithelial-mesenchymal transition, apoptosis, and cell migration by directly blocking its target, YWHAZ or ErbB2 in HCC cells (Li, Jia & Ding, 2018; Zhao et al., 2018). A potent oncogenic cooperation between mutant TP53 and ErbB2 was also demonstrated in human Her2-positive breast cancer (Li & Marchenko, 2017). On the basis of our findings, we reasoned that, in association with miR-221-3p and miR-375, TP53 could act as a tumor suppressor by inhibiting HBV replication, proliferation, migration, and invasion of HCC cells, and promote p53-mediated cell apoptosis.

Unfortunately, there are some limitations in this study. First, despite the fact that the unvalidated microRNAs had a similar numbers of targets to the validated microRNAs, only the two miRNAs with significant variation were selected and further analyzed. Moreover, a dual-luciferase reporter experiment was not performed to directly verify whether TP53 was the target gene of miR-221-3p and miR-375. The latter aspect will be the focus of forthcoming research in our laboratory.

Conclusions

In sum, we demonstrated that the aberrant expression of miR-221-3p and miR-375 may affect HBV-DNA replication, as well as the proliferation, migration, and invasion of HCC cells, possibly by modulating the expression of TP53. Thus, these two miRNAs may serve as crucial biomarkers for HCC diagnosis and prognosis.

Supplemental Information

Supplemental Information 1 Primers sequences for qRT-PCR.

Click here for additional data file.

Supplemental Information 2 GO and KEGG enrichment analysis of target gene sets predicted by three upregulated and downregulated miRNAs.

The top ten enriched (A) biological process, (C) molecular function, (E) cellular component, and (G) KEGG pathways analysis of target gene sets predicted by three upregulated miRNAs ranked by –log10(P-value); the top ten enriched (B) biological process, (D) molecular function, (F) cellular component, and (H) KEGG pathways analysis of target gene sets predicted by three downregulated miRNAs ranked by –log10(P-value). Red represented the –log10(P-value) of each cluster, blue represented enriched gene number of each cluster.

Click here for additional data file.

Supplemental Information 3 Raw data and images.

Figure data and images of WB, wound heal assay, and transwell assay

Click here for additional data file.

I would like to thank professor Xin-guang Liu for his valuable suggestions on the design of the present study.

Additional Information and Declarations

Competing Interests

Author Contributions

Human Ethics

Data Availability

The authors declare that they have no competing interests.

Yanming Liu conceived and designed the experiments, performed the experiments, analyzed the data, prepared figures and/or tables, authored or reviewed drafts of the paper, and approved the final draft.

Yue Cao performed the experiments, analyzed the data, prepared figures and/or tables, authored or reviewed drafts of the paper, and approved the final draft.

Wencan Cai analyzed the data, prepared figures and/or tables, authored or reviewed drafts of the paper, and approved the final draft.

Liangyin Wu analyzed the data, prepared figures and/or tables, authored or reviewed drafts of the paper, and approved the final draft.

Pingsen Zhao conceived and designed the experiments, analyzed the data, authored or reviewed drafts of the paper, and approved the final draft.

Xin-guang Liu conceived and designed the experiments, analyzed the data, prepared figures and/or tables, authored or reviewed drafts of the paper, and approved the final draft.

The following information was supplied relating to ethical approvals (i.e., approving body and any reference numbers):

YueBei People’s Hospital granted Ethical approval to carry out the study within its facilities (sumc-irb-2017).

The following information was supplied regarding data availability:

The raw measurements are available in the Supplemental Files.

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
