# Peer review of "Aberrant expression of two miRNAs promotes proliferation, hepatitis B virus amplification, migration and invasion of hepatocellular carcinoma cells: evidence from bioinformatic analysis and experimental validation"

_PeerJ, doi:10.7717/peerj.9100_

## Round 0.1 · original submission · Major Revisions

Please address critiques of all reviewers and amend your manuscript accordingly.

·

Basic reporting

1. Another round of language editing and revision might be required.
2. Line 76. The listed conditions are risk factors and not causes of HCC. Also there are no references to support the statement.
3. The authors calculated the adjusted p-values to account for multiple testing but they chose p-value < 0.05 along with fold-change as criteria for significance! A similar cutoff was used to select significant functional categories with no attempt at controlling for multiple testing.
4. Only the platform/server for performing the differential expression and enrichment analysis are mentioned in the text, GEO2R and Enricher. The exact method for performing the analysis on these platforms should also be mentioned. In addition, the method for standardizing probe intensities (Line 202) and co-expression analysis (Line 290) are not mentioned at all.
5. Figure S1. It is not clear what the bars show or what the colors mean. The figure legend and axes labels are also missing.

Experimental design

1. The choices to use top n features either in the differential expression, to construct PPI networks, to filter hub genes or to perform enrichment analysis are arbitrary. The authors may present the reasoning behind this filtering or show that these decisions improves the results.
2. Being a known target of differentially expressed microRNA doesn’t necessarily mean the gene itself is regulated, up or down. The reasoning for constructing regulated lists (Line 128) or the PPI network (Line 135) is not sound, or at least misapplied. These lists would be valuable to explore where the known target come from, but the wouldn’t directly indicate whether the categories (gene sets) that they come from are functional or not in the HCC context.
3. miR-222-3p has almost as many targets in the “hub genes” as miR-221-3p including 4 shared targets, but it was dropped from the analysis completely. Please, justify.

Validity of the findings

The study demonstrated the effect of two microRNAs and some of their target genes on the HCC proliferation and invasion. That much is clear, however, several analysis choices were taken to arrive at the importance of those two features in the first place. The choices were not sufficiently explained or justified.

Reviewer 2 ·

Basic reporting

n/a

Experimental design

n/a

Validity of the findings

n/a

Additional comments

In this study by Liu et. al. have reported that they have identified several up- and down-regulated mi-RNAs in HCC tumors. The authors focused on two miRNAs (miR-221-3p and miR-375) and claimed that they promoted proliferation, HBV amplification, migration, and invasion of HCC.

The involvement of miR-221 in HCC has been studied before and many of the conclusions of this paper were previously reported. For example, Fu Y et. al. (2019) have already shown that miR-221 promoted HCC by targetting PHF2, which in turn is a known regulator of p53 (Lee et.al., 2015). The authors failed to mention this study completely in this manuscript. Here are just a few peer-reviewed studies that have reported the involvement and effects of miR-221 in HCC:

1. https://www.ncbi.nlm.nih.gov/pubmed/31214616

2. https://www.ncbi.nlm.nih.gov/pubmed/31069760

3. https://www.ncbi.nlm.nih.gov/pubmed/30370582

4. https://www.ncbi.nlm.nih.gov/pubmed/31698701

5. https://www.ncbi.nlm.nih.gov/pubmed/31258758

Now the other miRNA that has been investigated in this report (miR-375) is also quite well known as a regulator of HCC proliferation. Upregulating miR-375 has shown to have an inhibiting effect in HCC by Lil L et. al. (2018). Here are a few among many other studies that looked at this relationship:

1. https://www.ncbi.nlm.nih.gov/pubmed/29858089

2. https://www.ncbi.nlm.nih.gov/pubmed/29555460

3. https://www.ncbi.nlm.nih.gov/pubmed/31681610

4. https://www.ncbi.nlm.nih.gov/pubmed/31406139

5. https://www.ncbi.nlm.nih.gov/pubmed/30302825

6. https://www.ncbi.nlm.nih.gov/pubmed/30127930

Could the authors comment on why this study is novel in light of these papers? Somehow authors have neglected to cite many of these papers which very clearly showed the involvement of miR-221 and miR-375 with HCC.

In my view, this manuscript has some novel aspects. However, this study suffers severely by failing to mention the already published literature and try to claim some already known facts as new findings. It is possible to review these (and many others) materials as the starting block and build a stronger paper which may include some of the experiments reported in this manuscript.

Reviewer 3 ·

Basic reporting

In the manuscript titled ‘Aberrant expression of two miRNAs promoted proliferation, HBV amplification, migration and invasion of hepatocellular carcinoma cells: Evidence from bioinformatic analysis and experimental validation’ the authors identify key miRNAs influencing the progression of hepatocellular carcinoma.

Although the manuscript is generally understandable, there are a lot of grammatical errors. I suggest authors proofread the manuscript. It might also be useful to have the manuscript reviewed by a native English speaker or edited through one of the English grammar software.
Example:
Line 90-91: the sentence is not well constructed.
During the last decades, Microarray technology have the advantages of high throughput, high sensitivity, and automatically.

Experimental design

Computational analysis and experiments are performed with proper controls. However, several important details are lacking in the Materials and Methods section.
Please rename ‘Materials and furthermore methods’ as ‘Materials and Methods’.
In several instances the authors provide ‘manufacturer’s instruction’ as the protocol for the experiment. However, the details of products/kits are not identified (other than the manufacturer’s name). Further details, such as the detailed name of the kit, catalog or a reference number available are necessary. Furthermore, provide a brief summary of the experimental procedure (if the detailed protocol is easily accessible). Also, cite other published work describing the protocol in detail.
Rewrite the entire Materials and Methods section to include appropriate details (both for computational work and the experimental work). This is necessary to make the data generation transparent and reproducible. For instance, not details have been provided regarding the source and nature of the antibody used, incubation conditions, and other details necessary to make this experiment reproducible.

Include the details of statistical analysis in the figure legend.
Please rewrite all the figure legends to make figures understandable and informative. In figure legend include:
1. The overall point/evidence the figure is providing
2. Provide the details what each figure panel is representing
3. Explain the observation
Example:
Figure 4B: The comparison of miR-221-3p expression between HCC tissues and matched paratumor tissues from 20 clinical samples indicate… (explain observation here).

Provide the details of the statistical test used in the figure legend.

Validity of the findings

It is critical that the appropriate details are provided in the Materials and Methods section to make the data acquisition transparent and reproducible.

---

## Round 0.2 · Minor Revisions

Please address the remaining issues indicated by the reviewer.

·

Basic reporting

no comment

Experimental design

no comment

Validity of the findings

no comment

Additional comments

The revised addressed some of the issues raised in the review. However, a few issues remain which could be addressed by the authors in the manuscript, if not by revising the analysis itself.

1. Regarding the adjustment for multiple testing, the specific technique used to calculate the adjusted p-values and the cutoffs need to be specified.
2. The authors explained that, on several occasions, they used the top n features to “limit the scope of target research.” This issue here is that the decision is completely arbitrary, this might result in loss of important signal. The authors may address this point in their discussion or employ an alternative approach. A data-driven decision might be a good solution for example by using object functions or natural divisions in the results.
3. The term “regulated lists” is misused as it should refer to features that are expressed differently between conditions, not just the targets of regulated microRNAs. Therefore, it would be beneficial if the authors rephrase this section of the manuscript. to reflect the reasoning they provide in the review.
4. Regarding choosing one microRNA to verify but not the other despite the fact that they have similar numbers of targets is understood. However, this needs to be presented as such in the manuscript, for example, as a limitation.

---

## Round 0.3 · accepted · Accept

Since all remaining critiques were adequately addressed and the manuscript was revised accordingly, the amended version is acceptable now.